# DSEE: Dually Sparsity-embedded Efficient Tuning of Pre-trained Language Models

## Abstract

Gigantic pre-trained models have become central to natural language processing (NLP), serving as the starting point for fine-tuning towards a range of downstream tasks. However, two pain points persist for this paradigm: (a) as the pre-trained models grow bigger (e.g., 175B parameters for GPT-3), even the fine-tuning process can be time-consuming and computationally expensive; (b) the fine-tuned model has the same size as its starting point by default, which is neither sensible due to its more specialized functionality, nor practical since many fine-tuned models will be deployed in resource-constrained environments. To address these pain points, we propose a framework for resource- and parameter-efficient fine-tuning by leveraging the sparsity prior in both weight updates and the final model weights. Our proposed framework, dubbed **D**ually **S**parsity-**E**mbedded **E**fficient Tuning (DSEE), aims to achieve two key objectives: (i) *parameter efficient fine-tuning* - by enforcing sparsity-aware weight updates on top of the pre-trained weights; and (ii) *resource-efficient inference* - by encouraging a sparse weight structure towards the final fine-tuned model. We leverage sparsity in these two directions by exploiting both unstructured and structural sparse patterns in pre-trained language models via magnitude-based pruning and $\ell_1$ sparse regularization. Extensive experiments and in-depth investigations, with diverse network backbones (i.e., BERT, GPT-2, and DeBERTa) on dozens of datasets, consistently demonstrate highly impressive parameter-/training-/inference-efficiency, while maintaining competitive downstream transfer performance. For instance, our DSEE-BERT obtains about $35\%$ inference FLOPs savings with $< 1\%$ trainable parameters and comparable performance to conventional fine-tuning. [1]

## 1 Introduction

Most recent NLP applications have been following the pre-train then fine-tune paradigm, where we start from a gigantic pre-trained model and fine-tune it towards downstream tasks. Conventional *fine-tuning* works by updating all of the parameters of the pre-trained model. As the size of pre-trained models grows, updating all parameters becomes less feasible in most practical scenarios, due to the expensive memory and computational requirements. For example, BERT$_{\text{BASE}}$ (Devlin et al., 2019) has 110M trainable parameters, while GPT-2 (Radford et al., 2019) has up to 1.5B and the largest version of GPT-3 (Radford et al., 2019) has an astonishing 175B trainable parameters. As such, conventional fine-tuning of the larger models could require hundreds of GPU hours. Another downside of this paradigm is that it requires storing as many parameters, as in the large-scale pre-trained models, for each downstream task. That poses impediments to the deployment in real-world resource-constrained environments.

One solution to address the extensive resource requirement of conventional fine-tuning is model pruning (LeCun et al., 1990; Han et al., 2015a; Ren et al., 2018; He et al., 2017; Liu et al., 2017), where unnecessary weights are eliminated to shrink the model size. For example, Chen et al. (2021b) leverages $\ell_1$ regularization to remove insignificant attention heads and gains $35\% \sim 45\%$ training time with comparable performance. Guo et al. (2020) learns sparse task-specific "diff" vectors for various downstream fine-tuning tasks, leading to great memory savings. All these studies indicate the rise of sparsity naturally during fine-tuning a general-purpose pre-trained model,

---

[1]All codes are provided in the supplement.

to some specialized downstream functionality. One potential interpretation, of why sparsity arises, is that different subsets of the parameters may be responsible for different downstream tasks and data domains (Sanh et al., 2020). However, identifying appropriate sparse pruning masks requires burdensome training of models with full weights, which can still be unaffordable for many practitioners. For example, fine-tuning a large pre-trained language model like GPT-3 for just one step consumes at least 1.2TB of VRAM and requires 96 pieces of NVIDIA Tesla (Hu et al., 2021).

One parallel alternative is designing *parameter-efficient* fine-tuning algorithms, which aims to only optimize a small portion of weights while fixing most of the pre-trained weights during the downstream task training step. Pioneering works along this line, which utilize adapters (Houlsby et al., 2019) or low-rank decomposition (Hu et al., 2021), can significantly reduce the number of trainable parameters while preserving good fine-tuning performance. Introducing only a small number of task-specific parameters for each new downstream task can substantially improve the memory and deployment efficiency of models as it allows us to reuse the remaining unchanged/shared parameters. However, there are two major hurdles of current parameter-efficient fine-tuning: ($i$) it does not yield any inference efficiency gains since the full pre-trained weights are still required to calculate outputs; and ($ii$) current methods assume the weight update to be either sparse (Guo et al., 2020) or low-rank (Hu et al., 2021), yet those assumptions might be oversimplified and overly restricted to allow for effective updates. For example, the low-rank assumptions on weight matrices might be overly strong since some weights cannot be fitted in the low-rank space (Yu et al., 2017). Recently Chen et al. (2021a) also find that using both sparse and low-rank components performs better than either of them. These observations have inspired us to explore better parameter-efficiency methods.

To tackle both resource- and parameter-efficiency issues of large model fine-tuning, we explicitly draw on the prior of sparsity for *both weight updates and the final weights*, and establish a *dually sparsity-embedding efficient tuning* (**DSEE**) framework. From a pre-trained model, DSEE first adopts a sparsity-aware low-rank weight update to achieve *parameter efficiency* of the fine-tuning process; and then enforces a sparse weight structure by masking to achieve *resource efficiency* of the fine-tuned model at inference time. Our contributions can be summarized as follows:

- We propose the dually sparsity-embedding efficient tuning (DSEE), which unifies sparsity-aware weight update and sparse pretrained weight in fine-tuning gigantic pre-trained models. It is the first attempt towards jointly optimizing both parameter efficiency of the fine-tuning process, and the resource efficiency of the fine-tuned model.

- We exploit both unstructured and structured sparse patterns in the DSEE framework. For weight updates, we find the injected sparsity prior to greatly enhance existing parameter-efficient update schemes, and to further trim down the needed amount of trainable parameters. For the final weights, we learn well-performed sparse masks from the pre-trained weights, leading to substantial computation and memory reductions at inference.

- Extensive experiments demonstrate the effectiveness of our proposal across various representative pre-trained languages models, such as BERT, GPT-2, and DeBERTa; and on diverse evaluation benchmarks, such as E2E, DART, WebNLG, and GLUE. Specifically, on (E2E, Dart, WebNLG), our methods can achieve a BLUE score of $(69.75, 55.40, 46.66)$ with less than 1% of total trainable parameters. On BERT, our method can save about 35% FLOPs, compared to traditional downstream fine-tuning.

## 2 RELATED WORK

**Pre-trained language models.** Transformer (Vaswani et al., 2017) is a sequence-to-sequence model that heavily uses the concept of self-attention. Later on, numerous transformer-based models have been proposed and show overwhelming performance on natural language processing (NLP) and on computer vision (CV) tasks. Devlin et al. (2019) designed BERT that pre-train deep bidirectional representations from unlabeled text and reached powerful performance. Liu et al. (2019) found that BERT were terribly undertrained and proposed RoBERTa, an enhanced training recipe for BERT which can greatly boost the performance. He et al. (2020) proposed decoding-enhanced BERT with disentangled attention (DeBERTa) that incorporates the disentangled attention mechanism and an improved mask encoder to enhance BERT and RoBERTa. More variants like XL-Net, Albert, and Electra have also been proposed in recent years (Yang et al., 2019; Lan et al., 2019; Clark et al., 2019). The series of GPT models (Radford et al., 2019; Brown et al., 2020) are later developed based on transformers decoder blocks rather than encoder blocks like BERT, which again have

shown superior performance on different tasks. These large models pretrained on a large amount of unlabelled texts would need to be further fine-tuned on downstream tasks for better performance. One of the accompanying disadvantages of these pre-training models with tremendous parameter counts (e.g., 175B in GPT-3) is the unaffordable computational cost for further fine-tuning.

**Pruning and Low-rank decomposition.** Pruning is a widely-used model compression technique. It can reduce the number of parameters inside models, which possibly brings training and inference efficiency. Along with weight pruning method (Han et al., 2015b) being one of the most effective methods (Gordon et al., 2020), various criterion have been proposed to select insignificant weights for pruning, such as Taylor approximation (Molchanov et al., 2019), Hessian score approximation (Hassibi & Stork, 1993), and other saliency scores such as SNIP (Lee et al., 2018), GraSP (Wang et al., 2019) and SynFlow (Tanaka et al., 2020). Several pruning methods have been commonly adapted to compress language models (McCarley et al., 2019; Gordon et al., 2020; Sanh et al., 2020; Wang et al., 2020; Chen et al., 2021b). Specifically, McCarley et al. (2019) proposed to prune attention heads that had less contribution to the model. Wang et al. (2020) pruned BERT models by involving low-rank factorization and $\ell_0$ regularization. Sanh et al. (2020) invented an improved version of magnitude pruning (*i.e.*, pruning based on the weight change) that can better suit the transfer learning. Chen et al. (2021b) performed structured pruning on BERT via $\ell_1$ sparse regularization, which reduced a large portion of parameters and decreased the training cost.

Low-rank approximation (Ye, 2005) is also vastly studied. One classical scenario is robust principle component analysis (Candès et al., 2011), which decomposes a matrix into a low-rank plus a sparse component. Existing literature shows that in deep learning, the learned over-parameterized models often naturally bear approximate low-rank weight structures (Oymak et al., 2019; Yu et al., 2017). Some (Jaderberg et al., 2014; Povey et al., 2018; Sainath et al., 2013; Zhang et al., 2014; Zhao et al., 2016) have explicitly imposed the low-rank constraint during training. Wang et al. (2020); Hu et al. (2021) utilized low-rank decomposition to shrink the model size and trim down the trainable parameters during fine-tuning. However, to our best knowledge, integrating sparsity and low-rank structures has never been studied before for efficient fine-tuning of pre-trained language models.

**Parameter-efficient adaptation.** Parameter-efficient adaptation aims at reducing the number of trainable parameters when fine-tuning the models across different downstream domains. Unlike pruning, it generates updates that can be represented by fewer parameters instead of building sparse models. Various approaches are invented to achieve the goal. Rebuffi et al. (2017); Houlsby et al. (2019) inserted and only trained adapters between existing layers, whose parameters are much less compared to the pretrained models. Guo et al. (2020) leveraged $\ell_0$ regularization to limit the number of non-zero elements in the update vectors. Lester et al. (2021); Li & Liang (2021) introduced efficient prompt tuning which optimizes only a small continuous task-specific vector. Hu et al. (2021) proposed a low-rank decomposition-based method that can also significantly reduce the number of trainable parameters. However, fine-tuned models yielded by these methods work have the same amount of weights as the pre-trained starting point; hence they contribute no resource efficiency of the final model.

## 3 METHODOLOGY

In this section, we begin by describing our notations and definitions of sparsity generation and parameter-efficient fine-tuning in Section 3.1. Then, we introduce the (dually) sparsity-embedded efficient fine-tuning algorithms in Sections 3.2 and 3.3.

### 3.1 PRELIMINARIES

**Sparsity generation and resource-efficient fine-tuning.** We adopt both unstructured and structured pruning methods to produce sparsity. They can lead to resource-efficiency including memory and computation savings.

Let $\mathcal{W} \in \mathbb{R}^{m \times n}$ denote a weight matrix. The goal of pruning is to find a binary mask $\mathcal{S} \in \{0,1\}^{\|\mathcal{W}\|_0}$, where $\|\mathcal{W}\|_0$ is the number of parameters in $\mathcal{W}$. The mask $\mathcal{S}$ is applied to $\mathcal{W}$ and results in a sparse weight $\mathcal{W} \odot \mathcal{S}$. For unstructured pruning, only memory cost is saved; but for structured pruning, it helps save computational cost since the sparse weights can be smaller in size by wiping out all-zero columns or rows. However, the performance of networks after structured pruning is often shown to be inferior compared with the unstructured pruning counterpart.

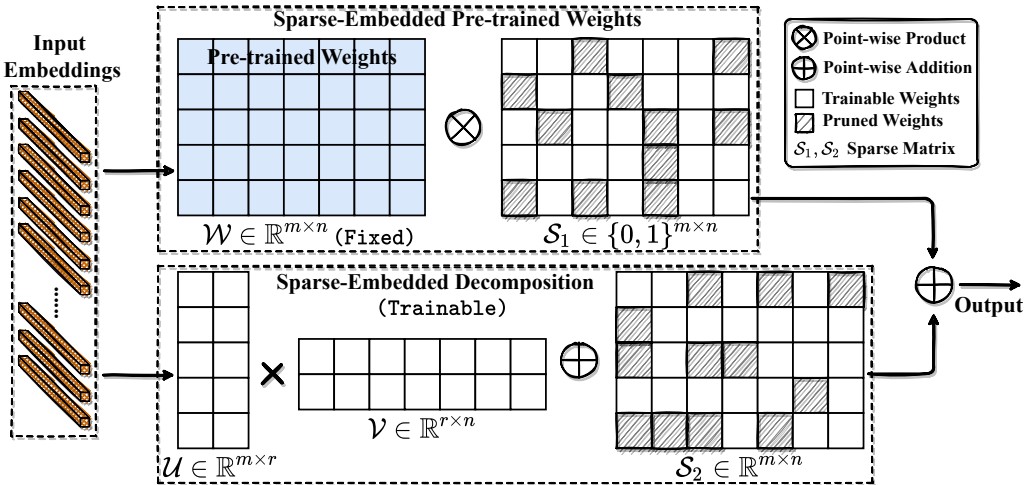

Figure 1: The overview of our proposals. The sparse masks can have unstructured or structured patterns, which leads to training and inference efficiency. During the fine-tuning, we only train decomposed matrices $\mathcal{U}$, $\mathcal{V}$ and non-zero elements in $\mathcal{S}_2$.

**Parameter-efficient fine-tuning.** To leverage the knowledge in pre-trained weights $\mathcal{W}$, downstream models learn task-specific weight update $\Delta\mathcal{W}$ via fine-tuning and generate predictions with weights $\mathcal{W} + \Delta\mathcal{W}$. The output of models is therefore calculated as $(\mathcal{W} + \Delta\mathcal{W})x$ where $x$ is the input. Since $\Delta\mathcal{W}$ has the same size of $\mathcal{W}$, learning the update matrices usually requires massive resources as the size of the pre-trained model increases. Parameter-efficient fine-tuning try to solve this problem by using as few trainable parameters as possible to represent $\Delta\mathcal{W}$, while maintaining competitive downstream fine-tuning performance. Previous literature reaches the goal via either sparsifying weight update matrices $\Delta\mathcal{W}$ (Guo et al., 2020) or leveraging low-rank decomposed matrices to compute $\Delta\mathcal{W}$ (Hu et al., 2021), while in our work we combine both of them.

---

**Algorithm 1:** Sparsity-Embedded Low-Rank Decomposition

**Input:** Pretrained weights $\mathcal{W}$, number of non-zero elements $N$

**Output:** Sparse matrices $\mathcal{S}_2$

1  Initialize $\mathcal{S}_2$ to be an empty set.
2  **for** *each self-attention projection weights $w_i$ in $\mathcal{W}$* **do**
   /* Decomposition          */
3     Perform matrix decomposition: $w_i \approx \mathcal{UV} + \mathcal{S}'$ by solving the optimization problem in Eqn.1.
   /* Identify important elements to form $\mathcal{S}_2$   */
4     Perform thresholding on $\mathcal{S}'$: Keep $N$ elements in $\mathcal{S}'$ with top magnitudes, and set the rest 0.
5     Append $\mathcal{S}'$ into $\mathcal{S}_2$.
6  **end**

---

**Algorithm 2:** DSEE

**Input:** Pretrained weights $\mathcal{W}$, number of non-zero elements $N$, desired sparsity $s$, loss function $\mathcal{L}$

**Output:** Sparse mask $\mathcal{S}_1$, matrices $\mathcal{U}, \mathcal{V}, \mathcal{S}_2$

1  Decompose $\mathcal{W}$ into $\mathcal{U}, \mathcal{V}$ and $\mathcal{S}_2$
2  (Re-)Initialization: $\mathcal{U} = 0, \mathcal{V} \sim \mathcal{N}(0, 0.02), \Omega =$ indexes of non-zero elements in $\mathcal{S}_2, \mathcal{S} = 0$
   /* I: train before pruning    */
3  Train $\mathcal{U}, \mathcal{V}, \mathcal{S}$ with respect to $\mathcal{L}$ under the constraint of $P_{\Omega^C}(\mathcal{S}) = 0$.
   /* II: pruning the model     */
4  Prune (1-s%) parameters in $\mathcal{W}$ globally by sorting the magnitude of $\mathcal{W} + \mathcal{UV} + \mathcal{S}$, deriving the sparsity mask $\mathcal{S}_1$
   /* III: tuning after pruning  */
5  Tune $\mathcal{U}, \mathcal{V}, \mathcal{S}_2$ for $E$ epochs for recovering the performance.

---

### 3.2 SPARSITY-EMBEDDED PARAMETER-EFFICIENT FINE-TUNING

A recent study (Hu et al., 2021) enforces low-rank constraint to weight update tensors $\Delta\mathcal{W}$, and obtains a satisfactory trade-off between parameter-efficiency and model quality. However, as revealed experimentally by (Yu et al., 2017), a part of the important information in the trained weights will also scatter outside the low-rank subspace, creating sparse "residuals". Inspired by these observations, we investigate a new sparsity-aware low-rank subspace of $\Delta\mathcal{W}$, and introduce the first component of our proposal in Figure 1, i.e., sparsity-embedded parameter-efficient fine-tuning.

Specifically, we identify a sparse subspace $\Omega$, that we can project our update matrices $\Delta\mathcal{W}$ to. The update matrix is then decomposed into two components: a low-rank part which is represented by the multiplication of two low-rank matrices $\mathcal{U} \in \mathbb{R}^{m \times r}$ and $\mathcal{V} \in \mathbb{R}^{r \times n}$, and a sparse residual

$$\mathcal{S}_2 = \mathcal{P}_\Omega(\Delta\mathcal{W}), \text{ where } \mathcal{P}_\Omega(\Delta\mathcal{W}) = \begin{cases} s_{i,j}, & (i,j) \in \Omega \\ 0, & (i,j) \in \Omega^{\mathrm{C}} \end{cases}, \ i = 1, 2, \ldots, m, \ j = 1, 2, \ldots, n.$$

As illustrated in Figure 1, the update matrix $\Delta\mathcal{W}$ can be approximated by $\mathcal{U}\mathcal{V} + \mathcal{S}_2$, in which the $\mathcal{U}$, $\mathcal{V}$, and the non-zero element of $\mathcal{S}_2$ are learnable parameters while $\Omega$ is fixed once determined. Compared to the full fine-tuning schemes has $m \times n$ individual trainable parameters, our method only has $(m+n) \times r + \mathrm{card}(\Omega)$. If $r$ is smaller than $\frac{m \times n - \mathrm{card}(\Omega)}{m+n} \lessapprox 0.5\min\{m,n\}$, which is a loose bound, our method is capable of substantially reducing trainable parameters for downstream fine-tuning. In practice, the value of $r$ very small compared to $m$ and $n$ so the savings are considerable.

The next question is how to find appropriate $\Omega$. Motivated by (Candès et al., 2011), we formulate our sparsity-aware decomposition as the following optimization problem:

$$\min_{\mathcal{U},\mathcal{V},\mathcal{S}_2} \ \frac{1}{2}\|\mathcal{W} - \mathcal{U}\mathcal{V} - \mathcal{S}_2\|_{\mathbf{F}}^2 \tag{1}$$
$$\text{s.t. } \mathrm{rank}(\mathcal{U}) \leq r, \ \mathrm{rank}(\mathcal{V}) \leq r, \ \mathrm{card}(\mathcal{S}_2) \leq c,$$

where $\mathrm{rank}(\cdot)$ indicates the rank and $\mathrm{card}(\cdot)$ indicates the cardinality of a matrix. Here we derive $\Omega$ from pre-trained weights $\mathcal{W}$, which is different from previous low-rank techniques for model compression since we perform prior-training decomposition and can not access $\Delta\mathcal{W}$ before fine-tuning. In this way, we actually assume that $\Delta\mathcal{W}$ shares a similar crucial subspace $\Omega$ with $\mathcal{W}$ (Sun et al., 2018). We use the GreBsmo algorithm (Zhou & Tao, 2013) to solve this optimization problem. This method adopts a similar idea as random projection and can solve the optimization problem rapidly. The details of the algorithm are in Appendix.

Algorithm 1 summarizes the detailed procedure of our proposal. We first decompose the pretrained weights $\mathcal{W}$ into three matrices: $\mathcal{U}$, $\mathcal{V}$ and $\mathcal{S}_2$. $\mathcal{U}$ and $\mathcal{V}$ are matrices of low-rank and $\mathcal{S}_2$ is a sparse matrix. After the decomposition, we re-initialize $\mathcal{U}$ and $\mathcal{V}$ to be of shape $\mathbb{R}^{m \times r}$ and $\mathbb{R}^{r \times n}$. The initial values for $\mathcal{U}$ are set to 0, and the initial values for $\mathcal{V}$ will follow a normal distribution as Hu et al. (2021) did. We do not use the decomposed value inside $\mathcal{S}_2$, but only the index of non-zero elements. We collect the indexes of non-zero elements into $\Omega$, and reset the values of $\mathcal{S}_2$ back to 0. During the training, we only update the elements of $\mathcal{S}_2$ whose indexes are in $\Omega$, as well as $\mathcal{U}$ and $\mathcal{V}$.

### 3.3 Dually Sparsity-Embedded Efficient Tuning (DSEE)

Besides parameter-efficiency, resource-efficiency is another major goal of our proposed framework. Sparsity-embedded low-rank weight updates alone do not necessarily lead to computational cost reductions since it works together with the dense pre-trained weights and the massive computation during forwarding processes is inevitable. To achieve both resource- and parameter-efficiency, we also promote a sparse weight structure to the final fine-tuned model, as demonstrated by the sparse mask $\mathcal{S}_1$ in Figure 1. Our DSEE explores both unstructured/structured sparsity patterns as follows.

$\triangleright$ **Pruning with unstructured sparse masks.** The unstructured sparse mask is the most widely used mask since it usually produces almost undamaged performance compared to its dense counterpart (Han et al., 2015a). It applies fine-grained manipulation on each individual model weight, while the generated irregular sparse masks bring limited hardware speedup (Wen et al., 2016). Specifically, based on the weight matrix $\mathcal{W}$, a sparse mask $\mathcal{S}_1$ can be calculated with various approaches, either heuristic or optimization-based. We adopt the one-shot magnitude pruning (Han et al., 2015a) in DSEE, due to its simplicity and competitive performance.

In our context, we create the sparse pattern $\mathcal{S}_1$ from $\mathcal{W} + \mathcal{U}\mathcal{V}$: First, it sorts the magnitude (i.e., absolute value) of individual weights and removes parameters with bottom $1 - k\%$ magnitude. Second, we further tune the value of the low-rank update matrices for a few epochs, which is important to keep the performance as stated in Han et al. (2015b). Third, the sparse mask $\mathcal{S}_1$ that we derive will be applied on the pretrained weights only and do not affect the output from the update matrices, as $\mathcal{W} \odot \mathcal{S}_1 + \mathcal{U}\mathcal{V}$. For an input sample $x$, the output is calculated by $(\mathcal{W} \odot \mathcal{S}_1 + \mathcal{U}\mathcal{V} + \mathcal{S}_2)x = \mathcal{W} \odot \mathcal{S}_1 x + (\mathcal{U}\mathcal{V} + \mathcal{S}_2)x$, where the first term $\mathcal{W} \odot \mathcal{S}_1 x$ brings significant resource-efficiency, and the second term brings parameter-efficiency.

▷ **Pruning with structured sparse masks.** The second method exploits structured sparse masks, whose regular patterns are more hardware friendly yet have worse performance than unstructured masks. Our DSEE considers $\ell_1$ sparse regularization (Liu et al., 2017; Chen et al., 2021b) to craft high-quality structurally pruned subnetworks. More precisely, motivated by Chen et al. (2021b), we introduce learnable coefficients $\xi$ before each attention head module, and append a $\ell_1$ sparse penalty on $\xi$ to the original loss. Detailed formulations are depicted as below:

We add trainable coefficients $c$ before attention heads. The parameters $c$ are optimized together with the decomposed matrices, *i.e.*, $\mathcal{U}, \mathcal{V}$ and $\mathcal{S}_2$. An extra term $\lambda\|c\|_1$ will be added to the training loss for sparse regularization. The value of $\lambda$ is set to 1e-4. After training, we prune the attention heads that have the lowest contribution to the model (*i.e.,* lowest $c$). We use a layer-wise pruning scheme that prunes the same proportion of heads in each attention layer. After pruning, several epochs of tuning are run to recover the performance of the pruned model. The size of update matrices will change after structured pruning, so we also need to change the dimension of $\mathcal{U}$ and $\mathcal{V}$. Specifically, we change the size of $\mathcal{V}$ from $\mathbb{R}^{r \times n}$ to $\mathbb{R}^{r \times [n \times s]}$ where $s$ is the pruning ratio of the corresponding self-attention layer and $[x]$ is the biggest integer not greater than $x$. The size of $\mathcal{S}_2$ is shrunk accordingly.

## 4 EXPERIMENT RESULTS

**Datasets and models.** We use three classical pre-trained language models in our experiments: BERT$_{\text{BASE}}$ (Devlin et al., 2019), GPT-2 (Radford et al., 2019), and DeBERTa-large (He et al., 2020), which have 12/24/24 layers with hidden size of 768/1024/1024 and 110/354/380M trainable parameters, respectively. For BERT and DeBERTa, we evaluate our method on the GLUE benchmarks (Wang et al., 2018). For GPT-2, we use E2E (Novikova et al., 2017), WebNLG (Gardent et al., 2017), and DART (Nan et al., 2021) for evaluations.

**Training and evaluation details.** For BERT and DeBERTa, we follow the default settings in Wolf et al. (2019); Devlin et al. (2019). We use the AdamW (Loshchilov & Hutter, 2017) optimizer for downstream training. The batch size for BERT and DeBERTa is 32, 8 per GPU. We train three epochs to search the sparse mask $\mathcal{S}_1$, and continue to tune the model for three epochs to converge. The initial learning rates are reported in A1, and we linearly decay them. For GPT-2, we follow the same hyperparameters as in Hu et al. (2021). We train the model for five epochs to search for the mask, and further tune the pruned model for two epochs.

**Evaluation Metrics.** For tasks on BERT and DeBERTa, we use the accuracy, Matthew's Correlation, and Pearson's r in the evaluation by default, which is also the conventional setting in the NLP community. On GPT-2, we use BLEU (Papineni et al., 2002), METEOR (Denkowski & Lavie, 2014), TER (Snover et al., 2006) and NIST (Doddington, 2002) as our evaluation metrics. To evaluate the efficiency of models, we use the number of trainable parameters to measure the parameter efficiency, use Sparsity in Pretrained Weights to measure the resource efficiency, and FLOPs to evaluate the computational efficiency. We add a star sign ($*$) to indicate the structured sparsity (*i.e.*, structurally prune the pretrained weights).

**Baselines.** On BERT and DeBERTa, we conduct comparisons with the following baseline methods: ❶ Fine-tune: directly fine-tunes the full model; ❷ EarlyBERT (Chen et al., 2021b); ❸ BERT Tickets (Chen et al., 2020); ❹ OMP: prunes the fine-tuned weights by magnitude, and fine-tune afterwards; and ❺ LoRA (Hu et al., 2021). We also report the Huggingface's fine-tuning results.

On GPT-2, we conduct comprehensive comparisons with multiple baseline methods: ❶ Adapters (Houlsby et al., 2019): insert adapters after linear layers; ❷ FT-Top2: fine-tune the top 2 layers only; ❸: Prefix: prefix tuning introduced by Li & Liang (2021); and ❹ LoRA: low-rank decomposition, which assumes $\Delta\mathcal{W} = \mathcal{U}\mathcal{V}$; most results are directly cited from Hu et al. (2021).

### 4.1 EFFICIENT TUNING WITH DSEE

**Parameter-efficiency with sparse masks.** To verify that using simple low-rank adaptation (*i.e.*, LoRA) has limitations, we compare its performance with the performance of our sparsity-embedded efficient fine-tuning. Table 1 proves that on MNLI, SST-2, CoLA, and STS-B, the simple decomposition form shows unsatisfactory results compared to sparsity-embedded decomposition. Adding a small proportion of parameters (only 3072 trainable parameters) can bring a performance boost to the model. Specifically, under

rank 8, the metrics increase (0.69/0.13/0.008/0.003) on SST-2/MNLI/CoLA/STS-B, respectively. Moreover, using only approximately half of the number of trainable parameters, our method can achieve comparable parameters compared to the state-of-the-art method LoRA. Table 2 shows the performance on GPT-2. For all three tasks, our method can achieve comparable results with only about half of the trainable parameters with LoRA with rank four and make substantial performance gain compared to LoRA with rank two. On WebNLG, our method even achieves a higher

Table 1: Performance comparison with BERT$_{\text{BASE}}$ on SST-2, MNLI, CoLA, and STS-B.

| Dataset | # Trainable Parameters | SST-2 | MNLI | CoLA | STS-B |
|---|---|---|---|---|---|
| Fine-tune | 109.9M | 92.32 | 82.12 | 0.570 | 0.890 |
| $\Delta\mathcal{W} = \mathcal{U}\mathcal{V}$ | 589.8K | 92.09 | 80.79 | 0.575 | 0.891 |
| $\Delta\mathcal{W} = \mathcal{U}\mathcal{V}$ | 294.9K | 92.09 | 80.49 | 0.580 | 0.892 |
| $\Delta\mathcal{W} = \mathcal{U}\mathcal{V} + \mathcal{S}_2$ | 298.0K | 92.78 | 80.64 | 0.588 | 0.895 |

BLEU score, 55.56 versus 55.29, with half of the trainable parameters. Such a phenomenon indicates that adding a sparse matrix to the low-rank decomposition can make substantial improvement.

Table 2: comparison of different decomposition on GPT-2. The formulas of decomposition are reported. Results of fine-tuning and the results of # Trainable Parameters = 0.39M are cited from Hu et al. (2021).

| Dataset | # Trainable Parameters | E2E | | | WebNLG | | | DART | | |
|---|---|---|---|---|---|---|---|---|---|---|
| Metric | - | BLEU | MET | NIST | BLEU | MET | TER | BLEU | MET | TER |
| Fine-tune | 354.92M | 68.2 | 46.2 | 8.62 | 47.60 | 0.39 | 0.50 | 46.0 | 0.39 | 0.46 |
| $\Delta\mathcal{W} = \mathcal{U}\mathcal{V}$ | 0.39M | 70.4 | 46.9 | 8.84 | 55.29 | 0.4143 | 0.3938 | 48.23 | 0.392 | 0.469 |
| $\Delta\mathcal{W} = \mathcal{U}\mathcal{V}$ | 0.20M | 69.2 | 45.9 | 8.74 | 55.23 | 0.4134 | 0.3957 | 46.49 | 0.387 | 0.477 |
| $\Delta\mathcal{W} = \mathcal{U}\mathcal{V} + \mathcal{S}_2$ | 0.20M | 69.8 | 46.5 | 8.79 | 55.56 | 0.4132 | 0.3916 | 47.08 | 0.390 | 0.472 |

**Resource- and parameter-efficiency with sparse masks.** We report the performance of DSEE, including both the number of trainable parameters and the sparsity in pretrained weights. We use an unstructured sparsity of 50%, and structured sparsity of 25% and 33%, which is equal to pruning 1/4 and 1/3 heads from each attention layer. For BERT$_{\text{BASE}}$ and DeBERTa-large, we set the $r$ (the low-rank dimension) to be 16, and $N$ (the number of non-zero elements in $\mathcal{S}_2$ ) to be 64. We also prune each of the intermediate layers using a structured sparsity of 40% as in (Chen et al., 2021b). For GPT-2, we set $r$ to be 2 in DSEE and 4 in LoRA. The choice of $N$ remains 64.

Table 3 summarizes the performance on BERT$_{\text{BASE}}$. On BERT$_{\text{BASE}}$, our DSEE method can: ❶ achieve parameter-efficiency at high level and retain the performance on various downstream tasks. On BERT$_{\text{BASE}}$, with only about 600K trainable parameters ($110/0.5929 \approx 200\times$ smaller than the full model), DSEE can achieve comparable performance on all GLUE tasks. ❷ DSEE can achieve resource-efficiency in the final models. Using either unstructured or structured sparse masks can reach our goal of reducing the number of parameters in the pretrained weights without sacrificing much performance. At the sparsity level of 50%, unstructured DSEE can achieve an accuracy of 90.46% on QNLI, surpassing the baseline of fine-tuning, whose result is 90.15%. At the sparsity level of 25%, the structured DSEE can have performance gains ranging from 0.46% to 2.22%, except for QQP and QNLI. At the sparsity level of 33%, the performance gains range from 0.11% to 2.04%, except for QQP and QNLI. ❸ Compared to other baselines, *e.g.,* EarlyBERT and BERT Tickets, our method benefits from parameter efficiency, which updates the weights more efficiently. These observations validate the effectiveness of our DSEE method.

We also calculate the inference FLOPs of BERT$_{\text{BASE}}$ on STS-B dataset. The original BERT$_{\text{BASE}}$ on STS-B takes FLOPs of $3.7835 \times 10^{14}$, while LoRA takes FLOPs of $3.8096 \times 10^{14}$, which is 0.69% higher. Conversely, the structured version of DSEE takes a FLOPs of $2.4921 \times 10^{14}$ at the sparsity of 25%, and $2.3867 \times 10^{14}$ at the sparsity of 33%, which is 34.61% and 37.38% lower than LoRA. This indicates that a large proportion of computational cost can be saved at inference phase.

Table 3: Performance comparison of different methods on BERT$_{\text{BASE}}$ on GLUE benchmarks. The star sign (*) in the sparsity column indicates that it represents structured sparsity.

| Methods | # Trainable Parameters | Sparsity in Pretrained Weights | CoLA | STS-B | MNLI | QQP | QNLI | MRPC | RTE | SST-2 |
|---|---|---|---|---|---|---|---|---|---|---|
| Fine-tune | 110M | 0% | 57.02 | 88.97 | 82.12 | 91.01 | 90.15 | 85.29 | 70.40 | 92.32 |
| EarlyBERT | 110M | 33% | 41.00 | - | 79.97 | 89.44 | 89.86 | 80.39 | 61.01 | 90.94 |
| BERT Tickets | 110M | {50%,50%,70%,90%,70%,50%,60%,60%} | 54.5 | 88.4 | 82.4 | 90.2 | 89.1 | 85.2 | 66.2 | 92.1 |
| OMP | ∼55M | 50% | 56.17 | 88.66 | 81.97 | 90.72 | 89.80 | 82.84 | 70.04 | 92.18 |
| LoRA | 589.8K | 0% | 58.58 | 89.10 | 80.79 | 86.43 | 88.16 | 86.27 | 71.48 | 92.09 |
| DSEE | 592.9K | 50% | 56.74 | 88.77 | 81.41 | 87.21 | 90.46 | 85.05 | 70.04 | 90.83 |
| DSEE | 592.9K | 25%* | 59.01 | 89.90 | 84.34 | 90.99 | 90.94 | 86.76 | 71.84 | 92.78 |
| DSEE | 592.9K | 33%* | 57.79 | 89.93 | 84.16 | 90.78 | 91.09 | 87.25 | 71.48 | 92.43 |

Table 4 summarizes the performance on GPT-2, which shares a similar trend. Unstructured DSEE can achieve $2000\times$ reduction in trainable parameters and $2\times$ reduction in the final fine-tuned model size with almost no loss in downstream task performance compared to conventional finetuning. When compared to LoRA, the unstructured DSEE can retain the same level of the number of trainable parameters and downstream task performance, while showing a $2\times$ reduction in the final fine-tuned model size. The structured DSEE on GPT-2 seems to be less competitive than on BERT, but it can still hold performance on E2E and WebNLG after pruning 25% of heads in attention modules.

Table 4: Performance comparison of different methods on GPT-2 on E2E, WebNLG and DART. LoRA[†]: reproduced results. The star sign (*) in the sparsity column indicates that it represents structured sparsity.

| Methods | # Trainable Parameters | Sparsity in Pretrained Weights | E2E | | | WebNLG | | | DART | | |
|---|---|---|---|---|---|---|---|---|---|---|---|
| Fine-tune[2] | 354.92M | 0% | 68.2 | 0.462 | 8.62 | 47.6 | 0.39 | 0.50 | 46.0 | 0.39 | 0.46 |
| Adapters | 11.48M | 0% | 68.9 | 0.461 | 8.71 | 55.2 | 0.41 | 0.39 | 45.4 | 0.38 | 0.46 |
| FT-Top2 | 25.19M | 0% | 68.1 | 0.460 | 8.59 | 33.5 | 0.26 | 0.75 | 38.1 | 0.34 | 0.56 |
| Prefix | 0.35M | 0% | 69.7 | 0.461 | 8.81 | 54.4 | 0.41 | 0.41 | 45.7 | 0.38 | 0.46 |
| LoRA | 0.39M | 0% | 70.4 | 0.468 | 8.85 | 55.3 | 0.41 | 0.39 | 47.5 | 0.39 | 0.45 |
| LoRA[†] | 0.39M | 0% | 70.06 | 0.467 | 8.84 | 55.29 | 0.4143 | 0.3938 | 48.23 | 0.392 | 0.469 |
| DSEE | 0.20M | 30% | 69.33 | 0.465 | 8.73 | 55.78 | 0.4163 | 0.3921 | 47.21 | 0.390 | 0.471 |
| DSEE | 0.20M | 50% | 69.75 | 0.469 | 8.78 | 55.40 | 0.4124 | 0.3935 | 46.66 | 0.389 | 0.471 |
| DSEE | 0.20M | 25%* | 69.48 | 0.464 | 8.75 | 54.64 | 0.4105 | 0.4030 | 26.94 | 0.247 | 0.727 |

Finally, we validate our method on De-BERTa. The results are displayed in Table 5. We use four datasets, CoLA, MNLI, MRPC, and RTE. They have greatly varied sizes, which are representatives of the GLUE benchmark. DeBERTa is a larger model, so applying low-rank decomposition with our

Table 5: Performance comparison of different methods on DeBERTa-large on CoLA, MNLI, MRPC and RTE.

| Methods | # Trainable Parameters | Sparsity in Pretrained Weights | Dataset | | | |
|---|---|---|---|---|---|---|
| | | | CoLA | MNLI | MRPC | RTE |
| LoRA | 786.4K | 0% | 63.36 | 88.90 | 90.44 | 75.09 |
| DSEE | 789.5K | 30% | 67.35 | 90.58 | 90.20 | 77.98 |
| DSEE | 789.5K | 50% | 63.82 | 90.03 | 89.71 | 74.73 |
| DSEE | 789.5K | 25%* | 63.62 | 89.93 | 89.96 | 75.24 |

hyperparameters cannot match the performance of fine-tuning. Compared to LoRA, our method reaches higher performance on downstream tasks, albeit it needs slightly more training epochs. However, such a slight extra cost for searching the sparse mask $\mathcal{S}_1$ can be amortized by the efficiency gained in the future inference, since the burden on resources such as storing the pretrained weights is relieved.

## 4.2 UNDERSTANDING DSEE

The position of embedded sparsity masks plays a crucial role in our proposed DSEE. For instance, applying sparse masks to the pre-trained weights or weight updates produces resource-and parameter-efficiency. Precisely, we compare four different methods on BERT$_{\text{BASE}}$:

Table 6: Performance comparison on BERT$_{\text{BASE}}$ with different masks embedded.

| Methods | # Trainable Parameters | Sparsity in Pretrained Weights | Dataset | | | |
|---|---|---|---|---|---|---|
| | | | SST-2 | MNLI | CoLA | STS-B |
| Fine-tune | 110M | 0% | 92.32 | 82.12 | 57.02 | 88.97 |
| $\mathcal{W} \odot \mathcal{S}_1$ | 55M | 50% | 91.28 | 81.97 | 0.562 | 0.887 |
| $\mathcal{W} \odot \mathcal{S}_1 + \mathcal{U}\mathcal{V}$ | 589.8K | 50% | 90.66 | 81.13 | 0.566 | 0.884 |
| $\mathcal{W} + \mathcal{U}\mathcal{V} + \mathcal{S}_2$ | 592.6K | 0% | 91.97 | 80.86 | 0.580 | 0.893 |
| $\mathcal{W} \odot \mathcal{S}_1 + \mathcal{U}\mathcal{V} + \mathcal{S}_2$ | 592.9K | 50% | 90.83 | 81.41 | 0.567 | 0.888 |

one-shot magnitude weight pruning ($\mathcal{W} \odot \mathcal{S}_1$), two DSEE's variants ($\mathcal{W} \odot \mathcal{S}_1 + \mathcal{U}\mathcal{V}$) and $\mathcal{W} + \mathcal{U}\mathcal{V} + \mathcal{S}_2$), DSEE ($\mathcal{W} \odot \mathcal{S}_1 + \mathcal{U}\mathcal{V} + \mathcal{S}_2$). The results are collected in Table 6. We can see that: ❶ NO embedded sparsity in the pretrained weights yields the overall best performance. This is intuitive since the valuable knowledge learned from massive pretraining data is intact and not removed. ❷ Embedding sparsity into the pretrained weights harms only little to no performance. A similar conclusion is drawn by Chen et al. (2020) which validated that a pruned model can have a similar performance as the dense model. ❸ Using sparsity-embed efficient fine-tuning with the sparse pre-trained weights can also preserve performance, as well as achieve parameter efficiency.

## 4.3 ABLATION AND VISUALIZATION

**Different methods to generate sparse matrix $\mathcal{S}_2$** In this section, we compare the performance of BERT$_{\text{BASE}}$ using different methods to generate the sparse matrix $\mathcal{S}_2$ and the corresponding space $\Omega$. Except for the aforementioned matrix decomposition method, we try to (1) randomly sample indexes into $\Omega$; and (2) directly select the indexes of elements of highest magnitude of $\mathcal{W}$ into $\Omega$. In Figure 2 we can see that using the matrix decomposition method has the highest metric overall. It is also noteworthy that sparse matrices generated by other methods will sometimes harm the performance.

Another factor is the number of non-zero elements inside $\mathcal{S}_2$. More non-zero elements in $\mathcal{S}_2$ reduce the parameter efficiency, while less non-zero elements increase the efficiency but may not be much beneficial to the accuracy or other metrics. Figure 2 also shows the relationship between number

of non-zero elements in $\mathcal{S}_2$ and the performance on fine-tuned model on SST-2. From the graph, we can see that using an $N$ of 64 seems to have the most stable results compared to other choices. Another important piece of information we can derive from the figure is that a higher number of non-zero elements in the $\mathcal{S}_2$ does not guarantee better performance.

**Ablation of # ranks.** The rank of low-rank decomposition $r$ is crucial to the transfer learning performance. A small $r$ will result in lower representation ability, and a large $r$ will bring more parameters and reduce the parameter efficiency. To find the best value, we conduct experiments with different ranks $r$ on four datasets, SST-2, MNLI, CoLA, and STS-B. The results are displayed in Figure 3. We add quadratic trend lines in the graph along with the discrete points to smooth

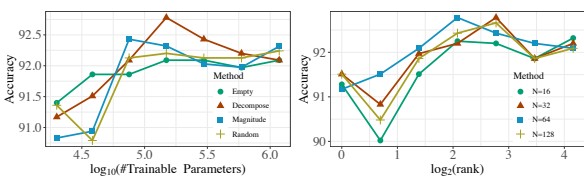

Figure 2: *Left*: Different method for generating $\mathcal{S}_2$ in BERT$_{\text{Base}}$. *Right*: Different number of non-zero elements in $\mathcal{S}_2$. Empty: no non-zero element in $\mathcal{S}_2$. Decompose: matrix decomposition method aforementioned. Magnitude: picking elements with highest magnitude. Random: random matrix.

the results. We can draw two conclusions from the graphs: ❶ Overall, the final performance is positively correlated with the number of trainable parameters; however, on some tasks (MNLI, CoLA) higher number of parameters (namely, the $r$ for $\mathcal{U}$ and $\mathcal{V}$) will lead to lower performance. ❷ With a sparse matrix embedded, the performance of the trained models can be improved within a range of trainable parameters. On SST-2 and CoLA, the applicable range seems to be $10^{4.5} \sim 10^6$. On STS-B, our method can consistently outperform the performance of using low-rank decomposition only. For MNLI, the two methods behave similarly while our method is slightly better.

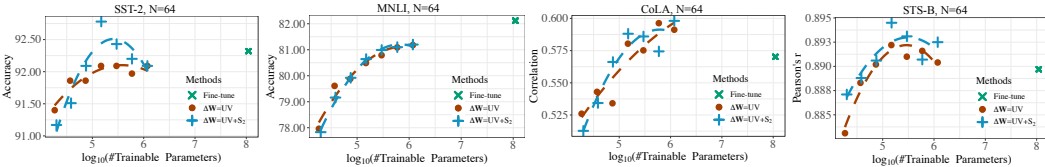

Figure 3: Performance comparison of two decomposition methods under different ranks. We add quadratic trend lines for better visualization quality.

**Ablation of sparsity.** Although creating sparsity in pre-trained weights $\mathcal{W}$ does not change the number of trainable parameters of models, the different levels of sparsity control the number of non-zero elements and thereby influence the representation ability. Therefore, we conduct experiments on BERT with different sparsity, ranging from 10% to 60%. The results are in Figure A5. From the figures, we can draw conclusions: ❶ DSEE out-performs magnitude pruning at low sparsity ($< 50\%$) with respect to the performance on downstream tasks. ❷ DSEE surpasses vanilla magnitude pruning with respect to the number of parameters. For each self-attention layer, using vanilla magnitude pruning needs to train all the weights, while DSEE only needs to update the three low-rank matrices.

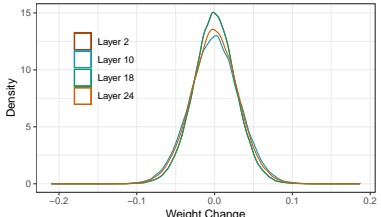

Figure 4: Weight change distributions.

**Visualizations.** Figure 4 shows the distribution of weight change. From the graph, we can see that most weights are located around 0. Such a phenomenon indicates that a natural sparsity exists within the update matrices, motivating us to explore sparse structures along with matrix decomposition.

## 5 CONCLUSION

This paper draws on the prior of sparsity and establishes the DSEE framework. It is the first attempt towards jointly optimizing both parameter efficiency of the fine-tuning process, and the resource efficiency of the fine-tuned model. On state-of-the-art large-scale language models (e.g., BERT, GPT, and DeBERTa) and across several of datasets, DSEE consistently demonstrates highly impressive parameter, training, and inference efficiency, in addition to preserving a competitive downstream transfer performance. Our future work targets extending DSEE to the finetuning of large-scale computer vision and/or multi-modal pre-trained models.

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

**All sections of newly added results and discussions are highlighted.**

## A1 MORE IMPLEMENTATION DETAILS

We report the setting of other hyperparameters of our experiments, such as learning rate, in Table A7. The device we used for experiments are various, including NVIDIA GeForce GTX 1080 Ti, GeForce RTX 2080 Ti, Titan RTX, and A6000.

| Architecture | Method | Parameters | Dataset | | | | | | | |
|---|---|---|---|---|---|---|---|---|---|---|
| | | | MNLI | QNLI | QQP | SST-2 | CoLA | MRPC | RTE | STS-B |
| BERT$_{BASE}$ | Fine-tune | Learning Rate | | | | | 5e-5 | | | |
| BERT$_{BASE}$ | DSEE (before pruning) | Learning Rate | 5e-5 | 5e-5 | 5e-5 | 2e-4 | 1e-3 | 8e-4 | 1e-3 | 8e-4 |
| BERT$_{BASE}$ | DSEE (after pruning) | Learning Rate | 5e-5 | 5e-5 | 5e-5 | 5e-5 | 1e-3 | 5e-4 | 5e-4 | 5e-4 |
| DeBERTa-large | LoRA & DSEE (before pruning) | Learning Rate | 1e-5 | - | - | - | 1e-3 | 8e-4 | 8e-5 | - |
| DeBERTa-large | DSEE (after pruning) | Learning Rate | 1e-5 | - | - | - | 5e-5 | 8e-4 | 6e-5 | - |

Table A7: Hyper-parameters we used on different datasets.

**GreBsmo Algorithm** GreBsmo (Zhou & Tao, 2013) is an algorithm for solving the Robust PCA-like methods. The optimization of $U$, $V$, and $S$ follows the following iterative rules:

$$\begin{cases} U_k = Q, \mathrm{QR}\left((X - S_{k-1})V_{k-1}^T\right) = QR \\ V_k = Q^T(X - S_{k-1}) \\ S_k = \mathcal{S}_\lambda(X - U_kV_k) \end{cases}, \tag{2}$$

where $X$ is the original dense matrix, $\mathrm{QR}(\cdot)$ means the QR decomposition, $\mathcal{S}_\lambda(\cdot)$ indicates the soft-threshold operation, and the subscripts $k$ indicates the optimization step.

## A2 MORE EXPERIMENT RESULTS

**Ablation of Sparsity** To study the relationship between sparsity of unstructured pruning and the behavior of our DSEE, we conduct an ablation study on various datasets in GLUE benchmarks. The results are shown in Figure A5

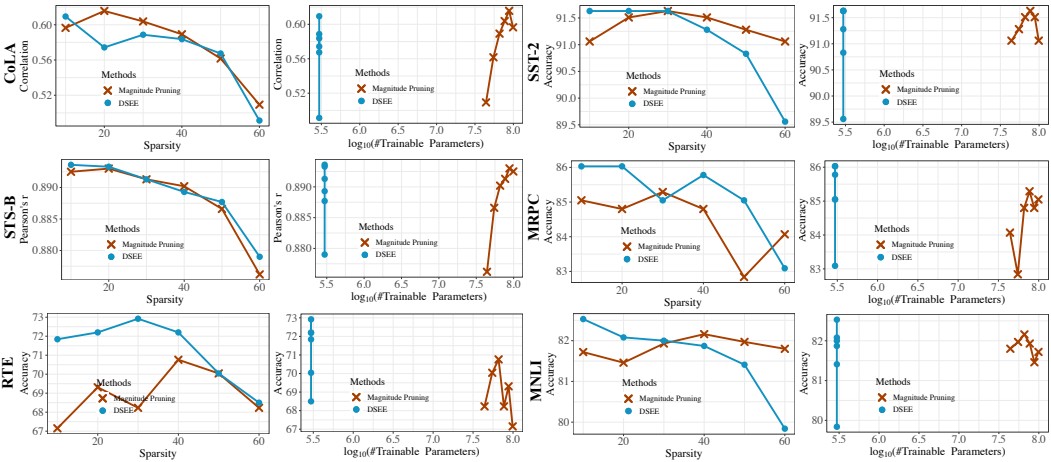

Figure A5: DSEE performance compared to vanilla magnitude pruning at different sparsity. Magnitude Pruning: vanilla magnitude pruning which tunes $\mathcal{W}$ directly.

**Results with more runs.** We conduct experiments with three runs. The mean and standard deviation of accuracy of LoRA and DSEE (without pruning) are shown in Table A8. We have also included the p-value of t-tests to show the significance of the performance gaps. On half of the

datasets, the p-value is around 0.1; and on three datasets the p-value is between 0.2 and 0.4. The reason why p-value is large on some datasets is probably because the number of experiments still has room to increase.

Table A8: Performance comparison of different methods on BERT$_{\text{BASE}}$ on GLUE benchmarks. The first row of each method indicates the mean accuracy, and the second row indicates the standard deviation.

| Methods | # Trainable Parameters | Sparsity in Pretrained Weights | Dataset | | | | | | | |
|---------|------------------------|--------------------------------|---------|-------|-------|-------|-------|-------|-------|-------|
| | | | CoLA | STS-B | MNLI | QQP | QNLI | MRPC | RTE | SST-2 |
| LoRA | 589.8K | 0% | 57.38 | 88.81 | 80.88 | 86.45 | 88.49 | 86.03 | 71.00 | 92.17 |
| | | | 1.43 | 0.23 | 0.41 | 0.14 | 0.10 | 1.49 | 1.10 | 0.27 |
| DSEE | 592.9K | 0% | 58.73 | 89.16 | 81.03 | 86.53 | 88.58 | 85.78 | 71.12 | 92.55 |
| | | | 0.95 | 0.03 | 0.62 | 0.06 | 0.17 | 1.07 | 0.62 | 0.12 |
| P-value of matched pair t-tests | | | 0.086 | 0.054 | 0.383 | 0.121 | 0.276 | 0.572 | 0.277 | 0.031 |
| DSEE | 592.9K | 50% | 55.60 | 88.39 | 81.16 | 87.23 | 88.88 | 84.88 | 70.09 | 91.00 |
| | | | 0.81 | 0.21 | 0.23 | 0.03 | 0.30 | 0.93 | 1.37 | 0.13 |

**Inference time on BERT$_{\text{BASE}}$**  We have recorded the inference time of BERT$_{\text{BASE}}$ on various GLUE benchmarks, QQP, STS-B, CoLA and RTE. The number of samples in their evaluation set range from 277 to 40430, so they can be considered representative. The results are shown in Table A9. From the table we can see that, the inference time are greatly saved after using the structured version of DSEE.

Table A9: Inference time of different methods on BERT$_{\text{BASE}}$ on GLUE benchmarks. The sparsity with star signs indicates that it is a structured sparsity.

| Dataset | # eval samples | Methods | | |
|---------|----------------|---------|------------------|------------------|
| | | LoRA | DSEE (25%$^*$) | DSEE (33%$^*$) |
| QQP | 40,430 | 50.773 | 36.391 | 34.946 |
| STS-B | 1,500 | 2.033 | 1.506 | 1.337 |
| CoLA | 1,043 | 1.316 | 0.931 | 0.877 |
| RTE | 277 | 0.464 | 0.420 | 0.376 |

**Convergence Speed**  We have compared the convergence speed (i.e., how the test accuracy changes) of LoRA and DSEE. From Figure A6 we can see that the convergence speeds of the two methods are not significantly different. This means introducing a sparse component into the decomposition form does not affect the convergence dynamics.

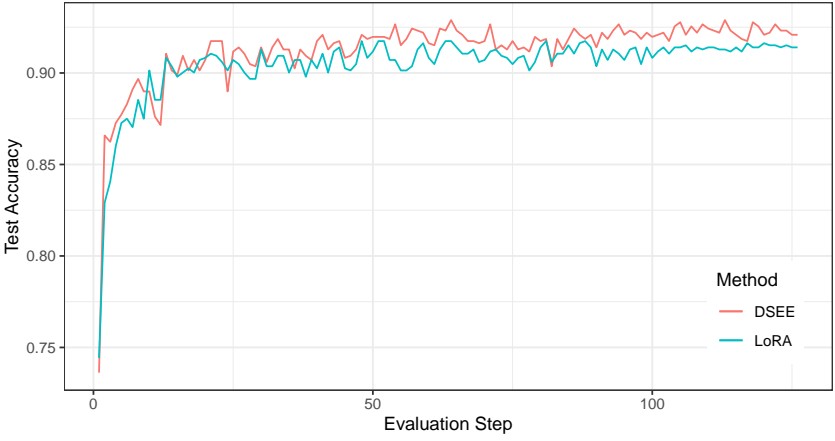

Figure A6: Convergence speed of two methods (LoRA and DSEE). The x-axis represents evaluation steps and y-axis represents the test accuracy.

