# OpenReview forum: "DSEE: Dually Sparsity-embedded Efficient Tuning of Pre-trained Language Models"
_ICLR.cc/2022/Conference — ICLR 2022 Submitted_

### Official Review · Reviewer_KvuY · 2021-10-31

**Correctness:** 3
**Technical Novelty And Significance:** 2
**Empirical Novelty And Significance:** 3
**Recommendation:** 5
**Confidence:** 4

**Main Review:**

The strengths and weaknesses are detailed below:

*Strengths:*

1. Combining weight-pruning and parameter-efficient tuning is a valuable attempt to further improve resource efficiency especially at inference time

2. Experiments demonstrate the effectiveness of the proposed method. The analysis sections regarding DSEE are also appreciated.

*Weaknesses:*

1. (This is a minor sanity check question) On GLUE tasks, are the results from one run or several random runs? This detail seems not to be mentioned in the paper – it is common practice to reports results from several random runs since the GLUE results are known to be sensitive to initialization

2. The necessity of adding a sparse matrix (i.e. S2 in the paper) into LoRA is not well supported. The justification experimental results in Table 1 and Table 2 are not very convincing to me because (1) the improvement from “+S2” is very small in most cases (and sometimes worse), for example, I do not think 0.008/0.003 are meaningful differences to claim in the paper; without significance test, most of the small differences may not be trustable; (2) In Table 1 and Table 2, many results from UV of 2x sizes (i.e. 589.8K in Table 1 and 0.39M from Table 2) are better than UV + S2. I know that this is not a fair comparison because of different models sizes, however, I suggest the authors double the size of UV + S2 for a fair comparison instead of downsizing UV – on the scale of 589.8K or 0.39M parameters, the number of parameters is not really a practical concern anymore. For example, your method can tune 1M less parameters than others, but such a 1M parameter saving may not lead to any meaningful differences in most practical settings.
Therefore, the sparse S2 adds complexity to the method and requires additional algorithm 1 in the pipeline, yet it does not yield very effective improvement as argued above, thus I do not think this is a proper design.

3. Without considering the questionable contributions of S2 because of point 2, I think the remaining technical contributions of this paper are limited because it is a combination of the existing parameter-efficient tuning method (LoRA) and existing weight pruning methods. Such a combination is empirically valuable but technically trivial.

4. The improvement of DSEE over LoRA seems from sparsity in pretrained weights, it is better to explain this phenomenon in the paper why weight pruning leads to a significant performance boost.

5. There is a sentence above Section 4.2 “DeBERTa is a larger model so applying low-rank decomposition with our hyperparameters cannot reach the performance of fine-tuning.” This statement is misleading and contradicts the claims by other papers [1] [2], which conclude that a larger model typically requires a lower-dimensional parameter update space.

6. The are some minor presentation flaws:
(1) In the first paragraph of the introduction section, citations to related papers should be included such as when mentioning BERT and GPT2.
(2) Above section 4.2, the authors say that Table 5 uses SST-2 …. which are not present in Table 5


[1] Lester et al. The Power of Scale for Parameter-Efficient Prompt Tuning. EMNLP 2021
[2] Aghajanyan et al. Intrinsic Dimensionality Explains the Effectiveness of Language Model Fine-Tuning. ACL 2021


**Summary Of The Paper:**

This paper proposes to enforce weight sparsity constraints into a prior parameter-efficient tuning method (LoRA) to achieve both parameter-efficient tuning and resource-efficient inference. Specifically,  this paper (1) adds another sparse, learnable weight matrix to the original low-rank decomposition in LoRA; (2) incorporates unstructured and structured weight pruning techniques to save computation at inference time. Experiments are conducted on GLUE and several data2text generation benchmarks to demonstrate the effectiveness of the proposed method.


**Summary Of The Review:**

Some core method design is not well-justified and the technical contributions are limited. There are some valuable empirical contributions though.

---

> ### Author Response · Authors · 2021-11-23
> **Response to Reviewer KvuY**
>
> Thank you for your valuable feedback! We have responded to your questions as below:
>
>
> **[Cons 1: Number of Runs and Significance of results and same-level experiments]**
>
> In the original version, we only run every experiment one time.  We have increased the number of runs to 3. We have also followed your instruction to include results from DSEE with the nearly-same level of trainable parameters.
>
> The results are shown below.
>
> |         Method         |     CoLA     |    STS-B     |     MNLI     |     QQP      |     QNLI     |     MRPC     |     RTE      |    SST-2     |
> | :--------------------: | :----------: | :----------: | :----------: | :----------: | :----------: | :----------: | :----------: | :----------: |
> |          LoRA          | 57.38 (1.43) | 88.81 (0.23) | 80.88 (0.41) | 86.45 (0.14) | 88.49 (0.10) | 86.03 (1.49) | 71.00 (1.10) | 92.17 (0.27) |
> | DSEE (without pruning) | 58.73 (0.95) | 89.16 (0.03) | 81.03 (0.62) | 86.53 (0.06) | 88.58 (0.17) | 85.78 (1.07) | 71.12 (0.62) | 92.25 (0.12) |
> |        p-value         |    0.086     |    0.054     |    0.383     |    0.121     |    0.276     |    0.572     |    0.277     |    0.031     |
> |       DSEE (50%)       | 55.60 (0.81) | 88.68 (0.21) | 81.16 (0.23) | 87.23 (0.03) | 88.88 (0.30) | 84.88 (0.93) | 70.09 (1.37) | 90.98 (0.13) |
>
>
>
> **[Cons 2: Proper Design and Technical Innovation?]**
>
> We respectfully disagree with you and believe that introducing S2 is meaningful:
>
> - The intuition has solid ground. The assumption of low-rank updates is too strong and limits the representation power of models. With the sparse component, the update matrices are allowed to occasionally “jump out” the low-rank space, which has stronger representation power. Such a phenomenon is also supported by recent research [r1].
> - The performance gain is significant as we can see from the results in the table above. With more runs, we can see that the p-values are overall significant.
>
> **[Cons 3: Why performance boost after pruning]**
>
> We think the reason why we are gaining performance after pruning is that the pruning serves as a regularizer. Pruning is proven to be effective in reducing network complexity and over-fitting [r1]. Therefore it is reasonable that after pruning the performance increases.
>
> [r1] Learning both Weights and Connections for Efficient Neural Networks
>
> **[Cons 4: Why DeBERTa cannot match]**
>
> It is noteworthy that in [r2] the intrinsic dimension is usually around hundreds, while we are operating at the highly aggressive low-rank dimension (16 and 8). The scales of dimension are so different and therefore the conclusion may not strictly hold in this small dimension range.
>
> [r2] Aghajanyan et al. Intrinsic Dimensionality Explains the Effectiveness of Language Model Fine-Tuning. ACL 2021
>
> **[Cons 5: Missing citations and results]**
>
> Thank you for pointing it out. We have revised these mistakes in the updated version of our paper.

---

> > ### Comment · Reviewer_KvuY · 2021-11-27
> > **Response to revision**
> >
> > Thank you for the response! The response and the revisions resolved some of the mentioned issues, and the paper is improved slightly with the added multiple random runs and revised presentations. However, my key concern -- the necessity of introducing S2 -- remains. While I agree that this design has good motivations, the empirical difference between DSEE (without pruning) and LoRA is indeed small, for example, <0.2 on 6 out of 8 datasets in GLUE on average across 3 runs from your updated results. The standard deviation is >0.1 in most cases, thus I do not consider the difference significant, your reported p-value is also quite large.
> >
> > I think my original comment holds:
> > > Therefore, the sparse S2 adds complexity to the method and requires additional algorithm 1 in the pipeline, yet it does not yield very effective improvement as argued above, thus I do not think this is a proper design.
> >
> > Therefore, I keep my original rating even though the paper is slightly improved on other aspects.

---

### Official Review · Reviewer_q6NF · 2021-11-08

**Correctness:** 2
**Technical Novelty And Significance:** 3
**Empirical Novelty And Significance:** 2
**Recommendation:** 5
**Confidence:** 4

**Main Review:**

This paper presents an interesting direction of achieving two different goals simultaneously. Below is my comments on this.

Strength:
1) Clearly defines the problem, goal and most part of the method
2) Although not fully novel to scratch, the present an interesting idea of achieving both parameter and resource efficiency. They uniquely studied the problem of this and presented a way of making it work.
3) Multiple solid experimental results with a diverse set of models, benchmarks, and metrics showing the generalization of their approach.

Weakness:
1) My major concern is the evaluation. It is difficult to combine the experiments to support the claims. The gains in performance are marginal in compare to LORA, where LORA is a simpler method. With the S1 matric in Table 6, it even became worse in comparison to LORA or the finetuned model. The main claim is to excel for the large models but as found, with large models like DeBERTA,  the gains in performance is even lower. The FLOPs results are extremely insufficient to stand as an evidence (only on SST-B are reported).
2) It's not described why S1 is only applied to pretrained weights W
3) Unclear when to use unstructured or structured sparsity.
4) Definitely just having a small reduction in memory usage does not mean to a significant resource utilization as it may lead to performance drop or may take an increased amount to training time. Analogously here, searching the sparsity also needs addition overhead (e.g., more epoch to tune UV after pruning etc.) the corresponding gain in performance is incremental specially when compare to the baseline LORA approach.

Typos and tips:
1) 1/3 heads from each attention head -> 1/3 heads from each attention leyer.
2) reset the values of S2 back to 0. During the update, we only update the elements of S2 whose indexes are in Ω -> this should be in correspondence to Algo-2.
3) More details how UV is computed without seeing pretrained weights W should be discussed (not just referring previous works)
4) FLOs on BERT -> FLOPs on BERT




**Summary Of The Paper:**

This paper proposes a principled way for achieving both parameter efficient finetuning and resource efficient inference. They combines the two steps of (i) computing a low-rank decomposition of parameter updates with a sparsity matrix (ii) further prune the trained (ie., finetuned) model. The first step aimed to achieve the parameter efficient finetuning but w/ the sparse residual components based on sparsity from the pretrained model to preserve the performance. The second step can reduce the memory of parameter storage. On three benchmarks with three different model architectures such as encoder-only, encoder-decoder they a large number of experiments with ablation studies. Studies show several a number of promising enhancement over other baselines.

**Summary Of The Review:**

Results with initial success but rejected as further evaluation is needed.

---

> ### Author Response · Authors · 2021-11-23
> **Response to Reviewer q6NF**
>
> Thank you for your valuable feedback! We have responded to your questions as below:
>
> **[Question 1: Marginal performance gain?]**
>
> We want to argue that our method can be directly compared with neither LoRA nor full fine-tuning since we introduce both parameter-efficiency and resource-efficiency into our method. One hurdle of LoRA is the requirement of pretrained weights - although the updates can be decomposed into low-rank components, the full pretrained weights are still involved in calculating the output of models. Our method, however, can reduce the size of the pre-trained weights, and therefore relieve the inference cost. Therefore the direct comparison within Table 6 is unfair.
>
>
> **[Question 2: Efficiency]**
>
> We have also conducted experiments to record the inference time using BERT on several datasets (QQP, STS-B, CoLA, and RTE).
>
> | Dataset | #samples | LoRA | DSEE (25%*) | DSEE (33%*) |
> | :------: | :---: | :-----------:| :------: | :------: |
> |  QQP   | 40430 | 50.773 | 36.391 | 34.946 |
> |  STS-B | 1500 | 2.033 | 1.506 | 1.337 |
> | CoLA | 1043 | 1.316 | 0.931 | 0.877 |
> | RTE | 277 | 0.464 | 0.420 | 0.376 |
>
> From the table, we can see that the structured version DSEE is capable of saving inference time on datasets with various sizes.
>
>
> **[Question 3: the position of S1?]**
>
> The introduction of S1 is to bring efficiency to the pre-trained weights. The sparse pre-trained weights have the potential to be accelerated, and the structured sparsity can bring a real reduction in FLOPs as shown in Table A9. The results can be also seen in the response to the last question.
>
> **[Question 4: when structured and when unstructured?]**
>
> Both methods have no requirement on application and can be applied under all scenarios. It is only based on users’ choices. Both methods have their own pros and cons - unstructured pruning hardly brings efficiency gain using the general device, while structured pruning may result in lower evaluation performance.
>
> **[Question 5: Overhead of searching]**
>
> Our method does not have a great overhead. The method for solving the problem defined in Eqn. 1 is based on matrix decomposition and can be done fast compared to the model fine-tuning. For example, the time for decomposition is 227 seconds for BERT, while the fine-tuning time is 1716 seconds. We have also recorded the convergence speed of both our method and LoRA in Figure A6. The results show that the convergence speed is unchanged after introducing the sparse component.
>
>
> **[Question 5: Minor Questions]**
>
> Thank you for pointing it out! We have revised and uploaded the paper. We have changed “heads” to “layers”, changed “FLOs” to “FLOPs”, revise the words describing the algorithms, and added an introduction on how U and V are computed.

---

> > ### Comment · Reviewer_q6NF · 2021-11-29
> > **Response to revision**
> >
> > Thank you for your response addressing my comments! I went through the the revisions, and yes it resolves some of the issues that I pointed out.
> >
> > However, I am still not yet convinced. In fact, although you reported the test time in A9, the results are missing. Even if I compile them from the main paper back and forth seems it has a very narrow advantages in compare to the baselines.
> >
> > Actually the results section is still very hard to conclude and the writing of the paper needs to go through some major revisions.
> >
> > Therefore, I do like the idea but I cannot recommend it in its current state.

---

### Official Review · Reviewer_UaPn · 2021-11-09

**Correctness:** 2
**Technical Novelty And Significance:** 1
**Empirical Novelty And Significance:** 1
**Recommendation:** 3
**Confidence:** 5

**Main Review:**

Strength:

1. The paper is presented in an easy to read manner.

2. I believe it's an important research topic to work on which can have a broad impact.

Weakness:

1. The organization can be improved. I feel the main motivation is actually in section 3 (page 4)

"However, as revealed experimentally by (Yu et al., 2017), a part of the important information in the trained weights will
also scatter outside......"

but authors didn't put it in the introduction. This makes me feel the method is not well motivated. And consequently I think the proposed method is a bit ad-hoc. There are many details of thinking not well explained so maybe it can be an interesting method but currently it reads to me that it's fairly incremental. I think to make me increase the score, the introduction should introduce more rationale of previous methods and why the proposed method can improve.

2. I believe the authors overlook an important work in the line of efficient fine-tuning (https://arxiv.org/abs/2101.00190). I didn't run the code myself but one of my collaborators told me this method in practice is fairly competitive without any effort of parameter tuning. I think the comparison to the work should be added.

3. I think listing #trainable is not a fair comparison. To best of my knowledge, storing the sparse coordinate information requires additional coding of the row/col of the scalar which is omitted by most sparse-pruning papers. But when it comes to comparison of low-rank methods, it will create an unfair situation as low-rank methods doesn't require additional memory usage so maybe other researchers doesn't put emphasis on this but I insisted the memory footprint information should be represented in the result tables.

4. Also, sparse method is in general slow without hardware accelerator. I think authors also know this as they wrote in the paper:

It applies fine-grained manipulation on each individual model weight, while the generated irregular sparse masks bring limited hardware speedup (Wen et al., 2016).

But somehow I didn't see any actions to cope with this and they just listed FLOP reduced which is again unfair to other structured methods such as structure pruning and low-rank. I think it's ok to require extra hardware accelerator. But to really benefit the research community, it's important also to present the weakness of method and I believe it's important to show the real wall-clock inference time instead of just the simulated #FLOP reduced. It's also interesting to know what's the wall-clock time saving in efficient fine-tuning part. As the low-rank method in general is easy to train and the proposed method seems to have many steps. So it seems to me that it's possible low-rank methods will have a bit more parameters but converges faster so the real time saving is better than the proposed method.

In fact, I think the empirical experiments doesn't really show a significant improvement over previous methods. For example, in Table 4 LoRA with  0.39M outperforms DSEE 0.17M and honestly 0.39M really doesn't increase too many parameters compared to DSEE. So I think the justified the efficacy of each step in DSEE and frankly present strength/weakness is rather important.

5. This leads to my concern that each functionality of the method is not well discussed. In particular, the ablation study of solving eq 1 looks weird to me. As an apparent way to solve eq 1 is via iterative methods. Fix S2, UV can be optimally decided by SVD and fixing UV S2 can also be determined easily. So to me I think iteratively solving UV and S2 is the best candidate to compare but it's not appearing in the paper. And we don't even know what's inside the GreBsmo algorithm as authors didn't explain it at all. That confused me a lot.

And if we look into the ablation study in Figure 2, there are some cases where other heuristics actually outperformed the DSEE. It's important to study why such cases happen but author didn't touch this part. I feel this then make the experiments not able to justify the proposed method.




**Summary Of The Paper:**

This paper tries to solve both efficient fine-tuning and model size compression simultaneously. It adopts the previous framework to model W and \delta W together and use different existing approaches to learn 2 parts in order.

**Summary Of The Review:**

Overall, I think the paper present an easily understandable method to an important and interesting question. But it reads to me many details are hidden or omitted so the efficacy of the proposed claim cannot be justified properly. I didn't list all of my concerns but at least the above mentioned five points should be addressed in order for me to increase the ratings.


Post rebuttal:

After author's reply, I still cannot confirm there is significant technical contribution. In addition, despite authors posted some additional experiment as requested, the numbers cannot match the original paper so I am not convinced the empirical result as well. In sum, there is no enough argument that the proposed method should work or any explanation why it should work. On the contrary, the experiments suggested that initilization plays important roles which might imply the claimed contribution might not be the real meat. Empirical evaluation also don't support the claim so I decide to remain my original ratings.

---

> ### Author Response · Authors · 2021-11-23
> **Response to Reviewer UaPn (Part 1)**
>
> We thank reviewer UaPn for the valuable feedback. We have made responses to the questions below:
>
> **[Cons 1: Motivation]**
> Thank you for your suggestion. We have made the introduction more clear and more motivated. We have added a paragraph explaining why we need sparse residuals in the decomposition form, and also why the proposed method can improve.
>
> We want to clarify that our work is not ad-hoc. The motivation of the proposed method is clear: the combination of low-rank and sparse components has a strong representation ability than only using the low-rank part. Such phenomenon is observed in multiple works [r1,r2], which is the ground of our motivation.
>
> [r1] Scatterbrain: Unifying Sparse and Low-rank Attention Approximation
> [r2] On Compressing Deep Models by Low Rank and Sparse Decomposition, CVPR 2020
>
> **[Cons 2: Missing Baseline? ]**
>
> We have reported the performance of Prefix-tuning on GPT-2 in Table 4. For convenience, we paste the results here:
>
> | Method | #trainable | E2E | WebNLG | DART |
> | :-----: | :------: | :---: | :-------: | :-----: |
> | Prefix-Tuning | 0.35M | 69.7 | 54.4 | 45.7 |
> | LoRA |  0.20M | 69.2 | 55.23 | 46.49 |
> | DSEE (without pruning) | 0.20M | 69.8 | 55.56 | 47.08 |
>
>
> Meanwhile, we want to clarify that our work does not solely focus on parameter efficiency but explores two types of efficiency at the same time. Our method has higher inference efficiency compared to Prefix-Tuning.
>
> **[Cons 3: Memory Footprints]**
>
> Thank you for your suggestion and we have added the memory taken by each method in the table below. When calculating the memory of sparse matrices, we adopt the COO format. The sparse masks are only required in the weight updates (we can directly set the pruned items in pretrained weights to 0), so if there are 64 non-zero elements in the sparse update component, the memory cost is no more than 64 * 4 * 3 = 768 bytes. This is a negligible memory cost compared to the total memory cost of large language models.
>
> **[Cons 4: Are sparse models always slow?]**
>
> We admit that the unstructured pruned models generally cannot be accelerated without special hardware/implementation, but for structured pruned models we can easily get acceleration since the layers are smaller. We explore both methods in our paper, which are shown in Tables 3 and 4. The structured pruned models can easily be accelerated at inference time.
>
> **[Cons 5: Is the comparison on FLOPs fair?]**
>
> We argue that our comparison with LoRA using FLOPs as metrics is fair. While we explore both unstructured and structured pruning in our paper, we only compare the FLOPs of the structured pruning with LoRA. Therefore, both the FLOPs reported (i.e., structured version of DSEE and LoRA) can truly act as indicators of efficiency since the reduction in FLOPs comes from the shrunk pretrained weights size (and not from parameters equal to 0).
>
> We have also followed your suggestion to include the inference time using BERT on several datasets (QQP, STS-B, CoLA, and RTE).
>
> | Dataset | #samples | LoRA | DSEE (25%*) | DSEE (33%*) |
> | :------: | :---: | :-----------:| :------: | :------: |
> |  QQP   | 40430 | 50.773 | 36.391 | 34.946 |
> |  STS-B | 1500 | 2.033 | 1.506 | 1.337 |
> | CoLA | 1043 | 1.316 | 0.931 | 0.877 |
> | RTE | 277 | 0.464 | 0.420 | 0.376 |
>
> From the table, we can see that the structured version DSEE is capable of saving inference time on datasets with various sizes.

---

> > ### Comment · Reviewer_UaPn · 2021-11-27
> > **Additional Question**
> >
> > Hi authors:
> >
> > Thanks for the response. Here I have additional questions for your reply.
> >
> > 1. Your results on Prefix-Tuning doesn't match the numbers in the original paper. Can you explain more on what entries you took from the paper? In addition, you mentioned prefix-tuning is using 0.35M but according to the original paper, they used 0.1% of total parameters which shouldn't end up this number. Can you also add more description on the setup?
> >
> > 2. On Cons 5 , what's the original inference time without approximation? What's the rank used in LoRA?
> >
> >
> > 3. Again, Cons 5, what's the inference time of an unstructured DSEE?
> >
> > 4. On [Cons 9: Performance in Figs]
> >
> > If your claim is due to randomness, then based on the results different initilizations are within an undistinguishable performance range so it reads to me that the proposed initilization doesn't contribute much. Do you have any explanation on this?

---

> ### Author Response · Authors · 2021-11-23
> **Response to Reviewer UaPn (Part 2)**
>
> **[Cons 6: Does the low-rank method converge faster than DSEE? Can the low-rank method save more time?]**
>
> We clarify that our method aims at reducing the inference cost and the number of trainable parameters at the same time, while LoRA cannot realize the former goal (inference efficiency). So the convergence speed is actually not a critical point here. However, we also compare the convergence speed as you requested. The results are shown in Figure A6. We can see that the convergence speed is almost the same.
>
> **[Cons 7: Does LoRA outperform DSEE?]**
>
> We want to argue that our method cannot be directly compared with LoRA since we introduce both parameter-efficiency and resource-efficiency into our method. One hurdle of LoRA is the requirement of pretrained weights - although the updates can be decomposed into low-rank components, the full pretrained weights are still involved in calculating the output of models. Our method, however, can reduce the size of the pre-trained weights, and therefore relieve the inference cost. Therefore the direct comparison within Table 4 is unfair.
>
> The BLEU from the fair comparison between our method and LoRA are shown below:
>
> | Method | E2E | WebNLG | DART |
> | :------: | :---: | :-----------:| :------: |
> |  LoRA  | 69.2 | 55.23 | 46.49 |
> | DSEE (without pruning) | 69.8 | 55.56 | 47.08 |
>
> The results of other metrics can be also found in Table 2. From the results on BLEU, we can see that our method can out-perform LoRA if not pruning the pre-trained weights.
>
> **[Cons 8: Baseline for ablations]**
>
> We have included a description of the GreBsmo algorithm in the updated version of our paper. The algorithm is actually in an iterative fashion - first derive U and V after fixing S, and then derive S with fixed U and V. Your suggestion is correct - and we have actually covered it. Using SVD for low-rank approximation “is provably optimal when constructed from SVD but the expensive time cost makes SVD prohibitive to large matrices” [r3], which is why we adopt an SVD-free algorithm here.
>
> **[Cons 9: Performance in Figs]**
>
> We have run more experiments to better confirm the performance gain of our method.  We have updated the figure in Table 2. The results (both accuracy and standard deviation) are shown below.
>
> | Method | Rank | Accuracy (sd) |
> | :------: | :---: | :-------------: |
> | Random | 2      | 91.466 (0.712) |
> | Magnitude | 2 | 91.470 (0.598) |
> | Decompose | 2 | 91.694 (0.447) |
> | Random | 4 | 92.030 (0.147) |
> | Magnitude | 4 | 92.042 (0.208) |
> | Decompose | 4 | 92.088 (0.182) |
> | Random | 8 | 92.108 (0.152) |
> | Magnitude | 8 | 92.147 (0.369) |
> | Decompose | 8 | 92.222 (0.302) |
> | Random | 16 | 92.278 (0.171) |
> | Magnitude | 16 | 92.395 (0.228) |
> | Decompose | 16 | 92.455 (0.164) |
>
> With more runs, we can see that our method can steadily outperform other heuristics. The reason why under a certain seed other heuristics can outperform our method is probably due to different random initializations.

---

### Official Review · Reviewer_gnKh · 2021-11-09

**Correctness:** 3
**Technical Novelty And Significance:** 2
**Empirical Novelty And Significance:** 3
**Recommendation:** 5
**Confidence:** 4

**Main Review:**

Strength:
* Developing more parameter-efficient and resource-efficient NLP models is important
* Experiments show promising results

Weakness:
* Language and presentation of the paper can be improved. For example, after reading the introduction, it is still quite confused for me to understand the method and motivation because of confusing word usages — what exactly is the relationship and difference between “sparsity-aware weight update” and “sparse final weight”?
* Experimental results can be better validated. Compared to LoRA, they claim the advantage of DSEE is that the “burden on resources such as storing the pertained weights are relieved”. To further validate this claim, it will be insightful to also compare with LoRA applied on sparse pertained model (i.e. first apply pruning on pertained model, then apply LoRA). There is some discussion on FLOPs in e.g. section 4.1, but it will be more clear to have a complete table on training/inference time and memory.
* Technical novelty is somewhat limited, or at least can be better justified.



**Summary Of The Paper:**

* This paper proposes a Dually Sparsity-Embedded Efficient Tuning (DSEE) which enforces sporty-aware weight updates on top of the pertained weights, and encourages a sparse weight structure towards the final fine-tuned model. On a diverse set of backbones and datasets, DSEE achieves 35% inference FLOPs savings with <0.1% trainable parameters without loss of accuracy.


**Summary Of The Review:**

This paper addresses an important problem and shows some promising results to improve the efficiency of NLP models. However, the presentation of the paper can be improved and there should be more justification of the motivation and novelty of the method. In particular, it seems that the main difference between this method and LoRA is the sparsity of pertained models, which is achieved mostly with existing techniques, yet it seems that LoRA (and/or similar method) can also be applied on top of sparse pre-trained models.

---

> ### Author Response · Authors · 2021-11-23
> **Response to Reviewer gnKh**
>
> Thank you for appreciating our motivation. Your comments are valuable for us to improve our paper. We have addressed your concerns below:
>
> **[Cons 1: Confusion in the introduction ]**
>
> We have rewritten and changed the structure of the article in the revised manuscripts. The term “sparsity-aware weight update” means sparsity-aware decomposition, i.e., decomposing the update matrices into low-rank and sparse components; and we have changed the term “sparse final weight” to become “sparse pretrained weights”. Some grammatical errors have also been corrected.
>
> **[Cons 2: LoRA applied on sparse pretrained network]**
>
> We take your suggestion and launch experiments to apply LoRA based on sparse pretrained networks. We use BERT and SST-2 as an example. We pick the magnitude pruning as the method to generate masks, and we set the pruning ratio to be 50% and 70%. The performance of LoRA applying on the sparse pre-trained network is 89.91% and 82.11%, which is nearly 1% and 4% lower.
>
>
>
>
>
> **[Cons 3: Tables for Inference Time]**
> Thank you for your suggestions. We have added a table in the Appendix to show the inference time. The results are displayed below at the same time.
>
> | Dataset | #samples | LoRA | DSEE (25%*) | DSEE (33%*) |
> | :------: | :---: | :-----------:| :------: | :------: |
> |  QQP   | 40430 | 50.773 | 36.391 | 34.946 |
> |  STS-B | 1500 | 2.033 | 1.506 | 1.337 |
> | CoLA | 1043 | 1.316 | 0.931 | 0.877 |
> | RTE | 277 | 0.464 | 0.420 | 0.376 |
>
> We can see that structured DSEE can substantially reduce the inference time.
>
>
> **[Cons 4: Technical Novelty]**
>
> We respectfully clarify that our work’s technical novelty is multi-folded:
>
> - Current methods assume the weight update to be either sparse (Guo et al., 2020) or low-rank (Hu et al., 2021), yet those assumptions might be oversimplified and overly restricted to allow for effective updates. We instead explore the combination of low-rank and sparse weight updates, which has stronger representation ability compared to either of them.
> - We for the first time propose resource- and parameter-efficient fine-tuning by leveraging the sparsity prior in both weight updates and the final model weights.

---

### Official Review · Reviewer_uLTd · 2021-11-11

**Correctness:** 3
**Technical Novelty And Significance:** 2
**Empirical Novelty And Significance:** 2
**Recommendation:** 3
**Confidence:** 3

**Main Review:**

Strengths:
- The topic of this paper is timely. Models are growing to immense sizes (at faster and faster rates), and the community is starting to come to a reckoning with their computational (and storage) budget. Reducing the impact both from a memory footprint, ability to fine-tune, and perform efficient inference can have immense impact.
- As an empirical result, some of the findings in this paper points towards the proposed method being worthwhile to try in practice.

Weaknesses:
- In general, the motivation and exact contributions of this work are hard to follow. In fact, I found it hard to understand what exactly the proposed method (and inference mechanism) is without first reading LoRA, the paper on which this work is based. For example, how outputs are even calculated are not explicitly spelled out (via equations) until Section 3.3. Personally, I also find the notation of $\mathcal{W} + \Delta\mathcal{W}$ to be slightly confusing when also talking in the context of "sparse updates" etc: it makes me think that $\Delta\mathcal{W}$ is a repeated gradient update of $\mathcal{W}$.

- It seems like a core deviation of this work from an ensemble of prior techniques is the addition of the sparse residuals. As such, it seems appropriate that this should be highlighted more/expanded on in the main introduction for this work. On this note, I'm still not that convinced by the motivation, particularly on identifying the subspace $\Omega$ from the pre-trained $\mathcal{W}$. Can you provide more explanation for why you assume $\Delta\mathcal{W}$ shares this subspace? Or why $\Omega$ isn't computed from the _sparse_ version of $\mathcal{W}$? Compared to random (Fig. 2), the performance gap seems fairly small, and is inconsistent across scales.

- It's hard to judge the practical benefit of the proposed approaches simply by looking at the number of trainable parameters and the number of flops. Actually exploiting these reductions to the full extent is hard (as the authors noted as well). It's also not really clear where the frontier is on total parameters vs. trainable parameters vs. accuracy is. Is it possible to give some real efficiency (i.e., wall clock) times? This would help shed some more light on the actual benefits vs. remaining challenges (how to actually implement this real-time).

- The benefit of $\mathcal{S}_2$ seems dubious. The differences are quite small, which makes me wonder if it is worth the extra complexity.

**Summary Of The Paper:**

This paper proposes methods to improve the efficiency of large pre-trained models (millions or billions of parameters) by combining multiple notions of _sparsity_ and/or _dimensionality reduction_ on the model weights. This is achieved via (1) a low-rank update matrix with fine-tunable parameters, and (2) pruning the original pre-trained weight matrix. These two sets of parameters are combined during inference. The authors validate their proposed method on several model architectures and datasets.

**Summary Of The Review:**

Overall, I think that the topic the paper tackles is important and timely. The paper proposes a compelling combination of existing techniques to increase various aspects of efficiency. That said, unfortunately, in its current version the paper is not so clear to follow (in motivation and implementation). In terms of technical novelty, it seems that the added components over LoRA might need a bit more work for their performance gains to be more convincing.

---

> ### Author Response · Authors · 2021-11-23
> **Response to Reviewer uLTd (Part 1)**
>
> We appreciate your feedback. We make further explanations to clarify the concerns as below.
>
> **[Cons 1: Motivation, exact contribution, and notions]**
>
> Thanks. We have put the methods in a more prioritized position in our paper. We have also added more descriptions of the notions to avoid confusion. For example, we have added the calculation method of the models’ output in Section 3.1, and we have changed “sparse updates” to “updates that can be represented by fewer parameters”.
>
> **[Cons 2: Core Deviation, and grounds of assumptions]**
>
> We respectfully disagree and argue that the sparsity in pre-trained weights is also a major deviation from previous works. Our method is not a direct ensemble of previous techniques but a framework for solving a new problem that considers both parameter- and resource efficiency.
>
> We would also explain in detail why we assume the ΔW share the same subspace as W here. Firstly, it has been shown in [LoRA] that the low-rank updates of weight matrix (ΔW) are mostly contained in the top singular directions of the weight matrix W. Therefore if we assume the existence of a sparse update component of (ΔW), it may not be in the singular directions of the weight matrix W (otherwise it would be contained in low-rank component). The assumption of sharing the same subspace then arises.
>
> We can definitely derive $\Omega$ from the sparse version of W. However we believe our choice is better: before we conduct pruning, we perform low-rank and sparse updates so that the magnitude pruning mask can be more accurate. If we decide to derive $\Omega$ from the sparse W, then we can only perform low-rank updates before pruning. There would result in a loss of precision of the sparse mask.
>
> To confirm the performance gain, we have launched more experiments with five random seeds (five runs). Due to the limited time windows in the rebuttal period, we only finish part of the experiments. The results are shown below:
>
> | Method | Rank | Accuracy (sd) |
> | :------: | :---: | :-------------: |
> | Random | 2      | 91.466 (0.712) |
> | Magnitude | 2 | 91.470 (0.598) |
> | Decompose | 2 | 91.694 (0.447) |
> | Random | 4 | 92.030 (0.147) |
> | Magnitude | 4 | 92.042 (0.208) |
> | Decompose | 4 | 92.088 (0.182) |
> | Random | 8 | 92.108 (0.152) |
> | Magnitude | 8 | 92.147 (0.369) |
> | Decompose | 8 | 92.222 (0.302) |
> | Random | 16 | 92.278 (0.171) |
> | Magnitude | 16 | 92.395 (0.228) |
> | Decompose | 16 | 92.455 (0.164) |
>
> From the results above we can see that our method can substantially outperform other heuristics and prove that the performance gain is significant. For example, the p-value of the t-test at rank 16 between our method and random is ​0.09756.
>
> **[Cons 3: Real efficiency]**
>
> Thank you for your suggestion. We have calculated the inference time on several datasets. We use QQP (40430 eval samples), STS-B (1500 eval samples), CoLA (1043 eval samples), and RTE (277 eval samples). They are of different scales, and can be seen as representatives.
>  and reported below:
>
> | Dataset | #samples | LoRA | DSEE (25%*) | DSEE (33%*) |
> | :------: | :---: | :-----------:| :------: | :------: |
> |  QQP   | 40430 | 50.773 | 36.391 | 34.946 |
> |  STS-B | 1500 | 2.033 | 1.506 | 1.337 |
> | CoLA | 1043 | 1.316 | 0.931 | 0.877 |
> | RTE | 277 | 0.464 | 0.420 | 0.376 |
>
> From the results above, we can see that the inference time can be substantially saved after applying structured DSEE.

---

> ### Author Response · Authors · 2021-11-23
> **Response to Reviewer uLTd (Part 2)**
>
> **[Cons 4: Benefits of S2]**
>
> We believe that the introduction of S2 has multiple benefits:
> - The intuition has solid ground. The assumption of low-rank updates is too strong and limits the representation power of models. With the sparse component, the update matrices are allowed to occasionally “jump out” the low-rank space, which has stronger representation power. Such a phenomenon is also supported by recent research [r1].
> - The performance gain is significant. We have launched experiments with more runs to confirm the performance gain after introducing S2. The results are listed below:
>
> |         Method         |     CoLA     |    STS-B     |     MNLI     |     QQP      |     QNLI     |     MRPC     |     RTE      |    SST-2     |
> | :--------------------: | :----------: | :----------: | :----------: | :----------: | :----------: | :----------: | :----------: | :----------: |
> |          LoRA          | 57.38 (1.43) | 88.81 (0.23) | 80.88 (0.41) | 86.45 (0.14) | 88.49 (0.10) | 86.03 (1.49) | 71.00 (1.10) | 92.17 (0.27) |
> | DSEE (without pruning) | 58.73 (0.95) | 89.16 (0.03) | 81.03 (0.62) | 86.53 (0.06) | 88.58 (0.17) | 85.78 (1.07) | 71.12 (0.62) | 92.25 (0.12) |
> |        p-value         |    0.086     |    0.054     |    0.383     |    0.121     |    0.276     |    0.572     |    0.277     |    0.031     |
>
> We can see that except on MRPC, the unpruned DSEE can clearly out-perform the LoRA, showing the practical benefit of introducing S2.
>
> - Finding and applying S2 only has little complexity. The decomposition can be efficiently done by algorithms such as GreBsmo and OptSpace. These methods include only matrix multiplication and QR decomposition and converge fast.  For example, the time for decomposition is 227 seconds for BERT, while the training time on SST-2 is 1716 seconds.
>
>
>
> [r1] Scatterbrain: Unifying Sparse and Low-rank Attention Approximation

---

> > ### Comment · Reviewer_uLTd · 2021-11-29
> > **Response to revision**
> >
> > Thank you for the detailed response to my comments! The revisions to the manuscript are helpful, and resolve some of the original issues. It is also easier to follow.
> >
> > That said, I am still not yet convinced of either the technical or practical advantages of this approach, and tend to agree with the other reviewers that the benefits of DSEE are not that obvious. While it may indeed provide some (statistically) significant improvements, the effect size is quite small, and too small to be of practical interest (considering the increased complexity of the design). For some of the other experiments, the p-values are quite high (e.g., >0.05) even though the test set sample sizes are fairly large, which also makes me question if some of the reported increases are due to chance.
> >
> > However, though I still would not yet recommend it for acceptance, I do like the idea and think that the paper has merit---and the potential to be a strong contribution. Overall, my prior summary still stands: the writing of the paper can be improved, and the benefits over the existing state of the art should be shown to be more practically significant.

---

### Decision · Program_Chairs · 2022-01-20

**Decision:**

Reject

**Comment:**

The paper integrates several dimensionality reduction and sparsity methods for improving the efficiency of large pre-trained models. Overall, the paper is interesting and discusses an important topic. However, it seems that it is not ready to be published at the current stage. I would encourage the authors to take reviewers comments into account and further improve the paper

The pros and cons of the papers are summarized in the following:

Pros:
+ Improving the efficiency of large pre-trained models is an essential research issue.
+ The idea is interesting although the technical novelty is a bit limited.

Cons:
- The key concern is that the technical and practical benefit of the proposed approach is not clear based on the results demonstrated in the experiments.
- The writing of the paper can be further improved in general to make the motivation more clear.